# Neoadjuvant ^177^Lutetium-PSMA-617 Radioligand Therapy for High-Risk Localized Prostate Cancer: Rationale, Early Clinical Evidence, and Future Directions

**DOI:** 10.3390/cancers17203330

**Published:** 2025-10-15

**Authors:** Whi-An Kwon, Jae Young Joung

**Affiliations:** 1Department of Urology, Hanyang University College of Medicine, Myongji Hospital, Goyang 10475, Republic of Korea; 2Research Institute of Geroscience and Precision Medicine, Myongji Medical Foundation, Goyang 10475, Republic of Korea; 3Department of Urology, Urological Cancer Center, National Cancer Center, Goyang 10408, Republic of Korea

**Keywords:** high-risk localized prostate cancer, immunogenic cell death, Lutetium-177, neoadjuvant radioligand therapy, prostate-specific membrane antigen, radioimmunotherapy, tumor immune microenvironment

## Abstract

**Simple Summary:**

High-risk localized prostate cancer (PCa) often behaves more like an early systemic disease than a confined lesion, frequently rendering local therapy alone insufficient. Neoadjuvant ^177^Lutetium-PSMA-617 radioligand therapy (RLT) offers a theranostic approach to target both the primary tumor and occult micrometastases before surgery. Early trials, including the LuTectomy study, demonstrate that the therapy is safe and can induce significant prostate-specific antigen declines and partial histologic responses; however, a single cycle rarely achieves a complete pathological response. Ongoing studies are now testing multi-cycle regimens and combinations with checkpoint blockade or androgen deprivation therapy to boost pathological complete response rates. Key hurdles include optimizing patient selection, defining surrogate endpoints such as metastasis-free survival, and balancing cost, logistics, and long-term safety—especially if potent alpha-emitters enter the clinic. However, if these challenges are met, neoadjuvant RLT could meaningfully improve cure rates for the most aggressive forms of localized PCa.

**Abstract:**

Men with high-risk localized prostate cancer (PCa) often have poor long-term outcomes, underscoring the need for improved neoadjuvant strategies beyond the current standard of care. Radioligand therapy with ^177^Lutetium-PSMA-617 (^177^Lu-PSMA-617) has emerged as a promising method to eliminate occult micrometastases while enhancing immune-mediated clearance of the primary tumor. Initial trials have affirmed the treatment’s feasibility and safety; however, they have consistently reported a lack of pathological complete response. This absence of profound initial tumor reduction necessitates further therapeutic advancements. The underlying rationale for future strategies is clear, as ^177^Lu-PSMA-617 promotes immunogenic cell death, potentially sensitizing immunologically “cold” tumors to checkpoint inhibitors. However, caution is warranted. The synergy observed between these therapies in advanced, metastatic castration-resistant PCa stems from a different biological context, and similar outcomes cannot be presumed in treatment-naïve, localized disease without rigorous validation. Continued progress hinges on developing improved metrics for success and patient selection. Simple prostate-specific antigen reductions have demonstrated minimal correlation with significant pathological outcomes in this setting, underscoring the critical need for validated surrogate endpoints and predictive biomarkers. Ultimately, large-scale randomized trials are essential to determine whether this investigational approach impacts key clinical outcomes—namely, metastasis-free and overall survival. While the strategy is theoretically sound, its capacity to enhance cure rates for high-risk localized PCa remains unverified.

## 1. Introduction

Prostate cancer (PCa) is the most prevalent non-dermatological malignancy among males and a leading cause of cancer-related mortality worldwide [1]. In 2024, an estimated 299,010 new cases are projected to be diagnosed in the United States, with 35,250 associated fatalities [1]. While the majority of men present with localized disease that tends to progress slowly and respond well to treatment, a significant fraction presents with high-risk features indicative of aggressive clinical behavior, metastatic progression, and, ultimately, cancer-specific mortality. For patients with high-risk localized PCa, standard local therapies, though administered with curative intent, often fail to achieve durable disease control [2].

Given this therapeutic limitation, there is an acute and unmet clinical need for innovative strategies that can address the systemic nature of high-risk localized PCa from the outset. The high rates of disease recurrence following even optimally delivered local therapies strongly suggest that, in many cases, high-risk localized PCa is not merely a localized condition, but an early systemic disease with occult micrometastases already present at diagnosis [1]. This insight reframes the core therapeutic challenge. The issue is not a fundamental deficit in local treatment modalities, such as radical prostatectomy (RP) or radiation therapy (RT), but rather a conceptual framework that inaccurately defines the disease as confined to the prostate. The focus, therefore, should shift from exclusively enhancing local control to achieving early and effective systemic management [3].

The integration of systemic therapies into the neoadjuvant setting, preceding definitive local treatment, represents a promising strategy to improve long-term outcomes for this vulnerable patient population [4]. Within the realm of innovative systemic therapies, ^177^Lutetium-PSMA-617 (^177^Lu-PSMA-617) radioligand therapy (RLT), which targets the prostate-specific membrane antigen (PSMA), has emerged as a compelling option [5]. This modality employs a “theranostic” principle, enabling the precise delivery of radiation to tumor cells throughout the body. In this comprehensive narrative review, we examine the landscape of high-risk localized PCa, explain the rationale for a neoadjuvant approach, detail the mechanism and evidence supporting ^177^Lu-PSMA-617 RLT, and scrutinize the emerging clinical data, immunomodulatory potential, and future challenges associated with advancing this potent therapeutic modality into a curative-intent setting.

## 2. Defining the High-Risk Patient and the Limits of Standard of Care

### 2.1. A Heterogeneous Definition: Comparing International Guidelines

The contemporary management of PCa relies heavily on risk assessment, a strategic approach that guides diagnosis, treatment selection, and follow-up recommendations [6]. Three key international bodies—the National Comprehensive Cancer Network (NCCN) in the United States, the European Association of Urology (EAU), and the American Urological Association (AUA) in partnership with the American Society for Radiation Oncology (ASTRO) and the Society of Urologic Oncology (SUO)—provide evidence-based guidelines for this risk-stratified approach. Although these guidelines converge on a foundational framework incorporating serum prostate-specific antigen (PSA) level, Gleason score stratified by Grade Group, and clinical T-stage, their specific definitions of high-risk disease exhibit subtle yet critical differences, as summarized in Table 1.

The 2025 NCCN guidelines [7] stratify advanced localized disease into two distinct categories: high-risk and very high-risk. A disease is classified as high-risk if it has one or more of the following features: a clinical T (cT) stage of cT3-cT4, Grade Group 4 or 5, or a PSA level > 20 ng/mL. The NCCN further designates a very high-risk subgroup for patients with at least two of the following: a clinical stage of cT3-cT4, Grade Group 4 or 5, or a PSA level > 40 ng/mL. This granular stratification acknowledges the markedly poor prognosis associated with particularly advanced local tumors or a combination of unfavorable clinical attributes.

The 2025 EAU guidelines [8] define high-risk localized PCa as having at least two of the following: a clinical stage of cT3–4, a Gleason score of 8–10, or a PSA level ≥ 40 ng/mL. Very high-risk PCa is defined as node-positive disease or, if node-negative, meeting the high-risk criteria mentioned above. The 2025 AUA/ASTRO/SUO guideline (last updated in 2022 and still current in 2025) [9] continues to use a three-tier risk system, designating localized PCa as high-risk when any one of the following is present: a clinical stage of cT3a, Grade Group 4 or 5 (Gleason 8–10), or a PSA level > 20 ng/mL. Unlike the NCCN, it does not create a separate “very high-risk” category; tumors with seminal vesicle invasion (cT3b), adjacent organ involvement (cT4), multiple high-risk features, or pelvic node positivity are classified as locally advanced and managed with intensified multimodality treatment rather than being labeled as very high-risk localized cancer.

The nuanced distinctions among these leading guidelines, particularly the explicit very high-risk classification by the NCCN, highlight a degree of clinical ambiguity. This absence of a single, universally accepted definition can complicate the interpretation and comparison of clinical trial outcomes and lead to discrepancies in patient management across different healthcare systems. Moreover, it reveals a pivotal concern: clinical assessments alone may inadequately capture the extensive biological heterogeneity inherent in aggressive PCa. This discrepancy underscores the urgent need for more sophisticated prognostic tools, such as molecular biomarkers or the pathological response to neoadjuvant therapy, to more accurately identify patients requiring treatment intensification [10].

It is crucial to acknowledge that these guideline definitions were historically based on conventional imaging (CT and bone scan). The increasing adoption of highly sensitive PSMA PET for initial staging, as demonstrated in the proPSMA trial [11], is actively reshaping this landscape. PSMA PET can identify previously occult nodal or distant metastatic disease, thereby upstaging a significant proportion of men initially classified as having high-risk localized disease [12]. This imaging evolution is refining the true ‘localized’ population, ensuring that patients enrolled in future neoadjuvant trials are more accurately staged and are the ones most likely to benefit from a strategy aimed at treating microscopic, rather than macroscopic, systemic disease [13].

### 2.2. The Sobering Reality of Recurrence and Mortality

For men diagnosed with high-risk localized PCa, the standard of care involves definitive local therapy, typically RP with extended pelvic lymph node dissection or external beam radiotherapy (EBRT) combined with long-term androgen deprivation therapy (ADT) [9]. While these treatments can be curative for some, a substantial proportion of patients experience disease recurrence, highlighting the limitations of a purely local approach for a disease that is often systemic at presentation.

The rates of treatment failure remain a major clinical challenge. Across various studies, between 27% and 53% of all patients undergoing RP or RT will eventually experience a rise in PSA level, a condition known as biochemical recurrence (BCR) [9]. For patients with the most severe disease features, the risk of recurrence and subsequent metastasis can exceed 50% [14]. This high failure rate strongly suggests that at the time of initial diagnosis and treatment, occult micrometastatic disease has already spread beyond the prostate and pelvic region. This renders local therapy alone insufficient for a cure, leading to significant long-term mortality. Even with aggressive local treatment, the 10-year prostate cancer-specific mortality (PCSM) for men in a high-risk group treated with RP was approximately 7.4% [8]. Other large series report 10-year PCSM rates for high-grade disease (Gleason score 8–10) ranging from 7% to as high as 15% after surgery [15]. For patients with cT3 disease, 10-year cancer-specific survival is approximately 85%, meaning 15% die from their cancer within a decade of treatment [16].

### 2.3. The Rationale for a Neoadjuvant Approach

The neoadjuvant strategy, which involves administering systemic therapy before definitive local treatment, is a well-established paradigm in other solid tumors, such as breast, esophageal, and rectal cancers [17]. Its application in high-risk localized PCa is based on several compelling theoretical advantages. The primary goal is the eradication of micrometastatic disease. Delivering a potent systemic agent before surgery or radiation therapy offers the opportunity to eliminate the disseminated tumor cells responsible for future relapse and mortality, addressing the fundamental limitation of local-only therapy.

Second, the neoadjuvant setting provides a unique “window of opportunity” for an in vivo assessment of treatment sensitivity. This concept is perhaps best illustrated by the experience in breast cancer, where the response of the primary tumor to neoadjuvant chemotherapy offers a powerful real-time biomarker of therapeutic efficacy [18]. A profound pathological response, particularly a pathological complete response (pCR) defined as the absence of any residual invasive tumor in the surgical specimen, is a strong surrogate for improved long-term survival [19]. The ability to observe this response transforms the primary tumor into a “living laboratory” for assessing drug efficacy. A poor response could identify patients at very high risk who may benefit from immediate adjuvant therapy or enrollment in clinical trials, while a strong response could justify de-escalating further treatment. This real-time biological feedback is a powerful tool for personalizing therapy and accelerating drug development—a principle adopted by regulatory bodies like the U.S. Food and Drug Administration (FDA) to expedite breast cancer drug approvals [20].

Third, neoadjuvant therapy can induce tumor downstaging. By shrinking the primary tumor and treating locoregional lymph node involvement, the therapy may increase the probability of achieving negative surgical margins—a key predictor of recurrence-free survival (RFS)—and facilitate nerve-sparing techniques, thereby improving functional outcomes [21].

Finally, this approach provides an invaluable platform for translational research. By analyzing pre-treatment biopsy and post-treatment surgical tissues, investigators can study the biological effects of a novel therapy on the tumor and its microenvironment, accelerating the discovery of predictive biomarkers and elucidating mechanisms of response and resistance. It is within this framework of addressing micrometastatic disease and gaining crucial biological insights that neoadjuvant ^177^Lu-PSMA-617 RLT is being explored as a transformative strategy for high-risk localized PCa [22,23]. Figure 1 provides a clear visual summary contrasting the standard treatment pathway with the proposed neoadjuvant RLT strategy, illustrating the latter’s aim to systemically target both the primary tumor and occult micrometastases prior to definitive local therapy.

It is critical to differentiate the rationale for neoadjuvant RLT from the historical experience with neoadjuvant ADT. While ADT successfully induces tumor downstaging, it has not demonstrated a significant benefit in metastasis-free survival (MFS) or overall survival (OS), likely due to its primarily cytostatic mechanism of action [24]. Neoadjuvant PSMA RLT, in contrast, offers a different, directly cytotoxic mechanism [25]. It must be emphasized that neoadjuvant RLT does not yet have data to support any benefit in MFS or OS [26,27]. The scientific rationale is the hypothesis that its cytotoxic nature and ability to target micrometastases could potentially translate into long-term survival benefits where ADT has failed. This hypothesis, however, remains unproven and is the central question for ongoing and future clinical trials [28]. This failure is likely rooted in its mechanism: ADT is primarily cytostatic, halting the proliferation of hormone-sensitive cells but exerting limited cytotoxic effect on the pre-existing androgen-independent clones and micrometastases that ultimately drive disease recurrence [29].

This failure is likely rooted in its mechanism: ADT is primarily cytostatic, halting the proliferation of hormone-sensitive cells but exerting limited cytotoxic effect on the pre-existing androgen-independent clones and micrometastases that ultimately drive disease recurrence [25]. This offers two key theoretical advantages over ADT: (1) It can eradicate both androgen-sensitive and -resistant tumor cells, provided they express the target, and (2) Its systemic administration allows it to treat not only the primary tumor but also the occult micrometastases responsible for the failure of local-only therapies [24,26]. It is this ability to treat microscopic disease with a curative-intent systemic therapy that underlies our hypothesis that neoadjuvant RLT may succeed where neoadjuvant ADT has failed.

## 3. The Theranostic Foundation: PSMA-Targeted RLT

The advent of PSMA-targeted RLT signifies a substantial advancement in PCa management [30], built upon the “theranostic” principle of using a single molecular target for both diagnostic imaging and personalized therapy. The success of ^177^Lu-PSMA-617 in the most advanced stages of the disease has provided a robust foundation for its investigation in earlier, curative-intent settings.

### 3.1. Mechanism of Action of ^177^Lu-PSMA-617

The therapeutic efficacy of ^177^Lu-PSMA-617 depends on the intricate interplay of its three core components: the target, the ligand, and the radionuclide [31]. The target, PSMA, is a transmembrane glycoprotein that is overexpressed by up to 1000-fold on the surface of PCa cells compared to benign prostate tissue [30]. Notably, its expression tends to increase with disease progression, reaching particularly high levels in poorly differentiated, metastatic, and castration-resistant tumors, making it an ideal target for intervention in aggressive disease states [32]. While PSMA-617 is the most clinically advanced ligand, leading to its regulatory approval, it is one of several small molecules developed for this purpose [25]. Other ligands, such as PSMA-I&T, are also in clinical development and function via a similar mechanism of high-affinity binding to PSMA and subsequent internalization of a radionuclide payload [27,32]. This review focuses primarily on ^177^Lu-PSMA-617 due to the wealth of data from pivotal trials, but the principles discussed are broadly applicable to this class of agents.

The ligand PSMA-617 is a small molecule specifically engineered for high affinity and specificity to the extracellular enzymatic site of the PSMA protein. Once the ligand binds, the PSMA-ligand complex is rapidly internalized by the cancer cell, trapping the attached radionuclide inside. The therapeutic payload is the radioisotope ^177^Lutetium (^177^Lu), which decays via β-particle emission [33]. These emitted β-particles, which are electrons, have an average path length of 0.67 mm and a maximum range of approximately 2 mm in soft tissue [34]. As they travel through tissue, they deposit energy, causing cellular damage, primarily to DNA. The radiation from ^177^Lu predominantly causes single-strand DNA breaks, which, if sufficiently numerous or left unrepaired, can lead to cell cycle arrest and apoptosis [35]. The millimeter-scale path length of these β-particles creates a beneficial “cross-fire” effect, whereby radiation from a targeted cell can also damage adjacent tumor cells that may have lower or no PSMA expression, thus helping to overcome some degree of target antigen heterogeneity [33,36,37].

This therapeutic agent is one half of a theranostic pair. For patient selection, the diagnostic agent PSMA-11 is radiolabeled with Gallium-68 (^68^Ga), creating 68Ga-PSMA-11. After intravenous administration, PET/CT imaging confirms adequate PSMA expression in the patient’s lesions, indicating that these lesions are likely to take up the therapeutic radioligand ^177^Lu-PSMA-617. This imaging-based selection process ensures that therapy is directed toward patients most likely to benefit [38]

### 3.2. Pivotal Evidence from the Metastatic Setting

The clinical adoption of ^177^Lu-PSMA-617 was driven by two landmark trials that established its efficacy in the most advanced and difficult-to-treat setting: metastatic castration-resistant prostate cancer (mCRPC) that has progressed after both novel hormonal agents and taxane-based chemotherapy [39]. These studies provide a high level of evidence for the efficacy of ^177^Lu-PSMA-617 in this patient population.

The VISION trial (NCT03511664) was a global, randomized, phase 3 study that enrolled 831 men with PSMA-positive mCRPC [25]. Patients were randomized 2:1 to receive ^177^Lu-PSMA-617 plus standard of care or standard of care alone. The results were practice-changing. The addition of ^177^Lu-PSMA-617 led to a significant improvement in the trial’s two primary endpoints: radiographic progression-free survival (rPFS) and OS. The median rPFS was more than doubled, from 3.4 months in the control arm to 8.7 months in the RLT arm, corresponding to a hazard ratio (HR) of 0.40 (95% confidence interval [CI], 0.29–0.57; *p* < 0.001). More importantly, ^177^Lu-PSMA-617 conferred a substantial OS benefit, extending the median OS from 11.3 months to 15.3 months (HR, 0.62; 95% CI, 0.52–0.74; *p* < 0.001).

The TheraP trial (NCT03392428), a randomized phase 2 study conducted in Australia, provided further compelling evidence for the efficacy of ^177^Lu-PSMA-617 [40]. This trial compared ^177^Lu-PSMA-617 directly against an active comparator, the chemotherapy agent cabazitaxel, in men with mCRPC who had progressed after docetaxel. Patients were rigorously selected using both PSMA-PET and FDG-PET to ensure high PSMA expression and exclude FDG-avid but PSMA-negative tumors. The trial met its primary endpoint, demonstrating a significantly higher PSA response rate (a decline of ≥50%) for ^177^Lu-PSMA-617 compared to cabazitaxel (66% vs. 37%). RLT also resulted in a longer PFS (HR 0.63; 95% CI, 0.46–0.86). With longer follow-up, OS was similar between the two arms. This result was heavily confounded by the high rate of crossover from the cabazitaxel arm to receive ^177^Lu-PSMA-617 upon progression, complicating the interpretation of the survival endpoint but not diminishing the clear evidence of RLT’s superior activity and lower toxicity [40].

The robust and consistent efficacy demonstrated in these trials, conducted in a patient population with very limited options and a poor prognosis, provides unequivocal proof of the agent’s potent anti-tumor activity. This success in the end-stage, palliative setting serves as the primary justification for investigating its use in earlier disease states. The logic is that if ^177^Lu-PSMA-617 is effective against highly resistant, widespread disease, it may be even more effective, and potentially curative, when applied to the lower-volume, less-treated micrometastatic disease characteristic of the high-risk localized PCa setting [41,42].

The success of ^177^Lu-PSMA-617 in the post-chemotherapy setting has propelled investigations into its use in earlier lines of therapy for metastatic disease. Landmark trials such as SPLASH (NCT04647526) and ECLIPSE (NCT05204927) are currently evaluating its efficacy against novel hormonal agents in the pre-chemotherapy mCRPC space. This clear trajectory toward earlier disease states provides a strong and logical impetus for the central question of this review: What is the potential role of ^177^Lu-PSMA-617-based strategies in the curative-intent, neoadjuvant setting for localized disease? While this review first establishes the foundational role of RLT as a monotherapy, it is the potential for this modality to serve as a backbone for combination strategies that may ultimately be required to impact long-term endpoints such as MFS and OS.

### 3.3. The Radiobiological Imperative: β- Versus Alpha-Emitters

The therapeutic effect of any RLT is dictated by the physical properties of the radionuclide it carries [37]. Understanding these properties is crucial for appreciating the current state of the field and its future direction, which increasingly involves exploring different types of radiation emitters. The two main classes of particles used in RLT are β- and alpha (α-) particles, which have fundamentally different radiobiological characteristics, as detailed in Table 2.

First, β-emitters, such as ^177^Lu, are electrons or positrons. They are characterized by a relatively low linear energy transfer (LET)—a measure of the energy deposited per unit of distance traveled through tissue—of approximately 0.2 keV/µm [43]. Because of this low energy deposition density, hundreds or even thousands of β-particle tracks, or “hits,” are required to pass through a cell’s nucleus to induce lethal damage. However, their longer path length in tissue (up to several millimeters) provides a valuable cross-fire effect, allowing them to treat larger tumors and overcome some degree of target antigen heterogeneity [44].

Second, α-emitters, such as Actinium-225 (^225^Ac) and Radium-223 (^223^Ra), are helium nuclei (two protons and two neutrons). They possess a very high LET of around 100 keV/µm, approximately 500 times greater than that of β-particles [45]. This dense ionization track is incredibly damaging to biological molecules. An α-particle can cause complex, irreparable double-strand DNA breaks, which are highly cytotoxic [46]. Consequently, as few as one to ten α-particle traversals through a cell nucleus can be sufficient to kill the cell [47]. This high potency is coupled with an extremely short range in tissue—typically less than 100 µm, or the span of just a few cell diameters [48]. This minimizes damage to nearby healthy tissues but also eliminates any significant cross-fire effect.

This distinction creates a strategic tradeoff in radiopharmaceutical design. There is no single “best” radionuclide; rather, the choice depends on the clinical objective [49]. For the neoadjuvant goal of debulking a macroscopic primary tumor, which is likely to be heterogeneous in its PSMA expression, the longer range and cross-fire effect of a β-emitter like ^177^Lu may be more beneficial [50]. However, for the goal of eradicating disseminated single tumor cells or microscopic clusters of cells (i.e., micrometastatic disease or minimal residual disease [MRD]), the extreme potency and short range of an α-emitter may be theoretically superior, delivering a lethal dose to isolated targets while maximally sparing surrounding normal tissue [51]. This understanding frames the evolution of RLT from a single-agent approach to a more sophisticated, tailored strategy where the choice of radionuclide could be based on the disease volume and the specific therapeutic intent.

**Table 2 cancers-17-03330-t002:** Radiobiological properties of clinically relevant radionuclides.

Radionuclide	Particle Emitted	Half-Life (days)	Max Energy (MeV)	Max Range in Tissue	Typical LET (keV/µm)
^177^Lu [52,53]	β	6.7	0.497	~2 mm	~0.2
^90^Y [54]	β	2.7	2.3	~11 mm	~0.2
^223^Ra [55]	α	11.4	5.0–7.5	40–100 µm	~80
^225^Ac [56,57]	α	9.9	5.0–8.4	40–100 µm	~100

Abbreviations: ^177^Lu, Lutetium-177; ^223^Ra, Radium-223; ^225^Ac, Actinium-225; ^90^Y, Yttrium-90; keV, kilo-electron volt; LET, Linear Energy Transfer; MeV, Mega-electron Volt; µm, micrometer.

## 4. Emerging Evidence for Neoadjuvant ^177^Lu-PSMA RLT

The transition of ^177^Lu-PSMA RLT from the palliative metastatic setting to the curative-intent neoadjuvant setting is recent, but rapidly advancing [58]. While large-scale phase 3 data are not yet available, a foundation of preclinical work and pioneering early-phase clinical trials are beginning to provide the first insights into the safety, feasibility, and potential efficacy of this approach.

### 4.1. Preclinical Rationale and Early Human Experience

The rationale for using ^177^Lu-PSMA agents in PCa treatment was established through extensive preclinical investigations. In vitro studies using PSMA-expressing cell lines confirmed the high binding affinity of various PSMA-targeting compounds. Subsequent studies using murine xenograft models of PCa demonstrated specific and high-level accumulation of radiolabeled PSMA ligands within tumors, resulting in significant, dose-dependent tumor growth inhibition and the induction of DNA damage [59]. These assessments provided the essential proof-of-concept for the delivery of PSMA-targeted radionuclides. The first human applications of ^177^Lu-PSMA RLT were initiated through compassionate use programs, primarily in Germany and Australia [5]. Although not conducted under the stringent protocols of a formal clinical trial, these efforts yielded critical preliminary data indicating that the therapy was generally well tolerated and capable of eliciting significant PSA responses and tumor regression in heavily pre-treated patients. This empirical evidence from real-world scenarios was pivotal in generating momentum and providing the safety justification needed to launch prospective clinical trials, which ultimately culminated in the landmark VISION and TheraP studies.

### 4.2. Initial Clinical Trial Data: The LuTectomy and Golan Studies

The current landscape of neoadjuvant RLT is defined by a small number of innovative, investigator-initiated trials. For example, the LuTectomy trial (NCT04430192) was a single-arm, phase 1/2 study in Australia designed as a first-in-human evaluation of neoadjuvant ^177^Lu-PSMA-617 [60]. The trial enrolled 20 men with high-risk localized PCa or locoregionally advanced PCa scheduled for RP and pelvic lymph node dissection. A key inclusion criterion was high PSMA avidity on a screening 68Ga-PSMA PET/CT scan. Participants received a single intravenous infusion of approximately 5 GBq of ^177^Lu-PSMA-617 six weeks before their planned surgery.

The initial results, presented at the 2023 EAU annual meeting, provided foundational insights into this approach [26]. Regarding safety, the treatment was well tolerated. The adverse effects were consistent with those observed in the metastatic setting, with mild xerostomia, transient nausea, and fatigue being the most common toxicities. Surgeons confirmed that the subsequent RP was safe and feasible, reporting no unexpected intraoperative difficulties or increased complications related to the prior RLT. This is a crucial finding, as it establishes the fundamental viability of combining neoadjuvant RLT with surgery.

The primary endpoint of absorbed radiation dose confirmed that the therapy delivers a high and targeted radiation dose to disease sites. The median absorbed dose to the prostate was 19.6 Gy, and to the involved pelvic lymph nodes was even higher at 37.9 Gy. While these doses are substantial, they are lower than those delivered with EBRT (typically > 70 Gy), and there was significant inter-patient variability in dose delivery.

The preliminary efficacy data were also encouraging. The single cycle of RLT induced a median PSA decline of 49%, with 45% of patients achieving a PSA50 response (a decline of ≥50%). Post-treatment PSMA PET scans revealed that 55% of patients had a partial response, defined as a decline in SUVmax of >30%, while 40% had stable disease. At a median follow-up of nearly 14 months, the biochemical RFS rate was 80%.

However, the most critical findings came from the pathological examination of the prostatectomy specimens. While 80% of patients showed evidence of a partial histologic response—including features such as stromal fibrosis, reduced tumor cell density, and cytoplasmic vacuolation—not a single patient achieved a pCR. One patient (5%) was found to have only MRD. This outcome, while demonstrating clear biological activity, is highly significant: it strongly suggests that a single cycle of ^177^Lu-PSMA-617 monotherapy is insufficient to completely eradicate macroscopic, high-risk localized PCa. This “negative” result is not a trial failure but rather a valuable, hypothesis-generating insight. It establishes the lower boundary of efficacy for this approach and provides a compelling rationale for developing strategies to enhance the anti-tumor effect, such as using multiple RLT cycles, dose escalation, or combination therapies [60,61].

Complementing these findings, a study from Israel by Golan et al. (NCT04432015) investigated neoadjuvant ^177^Lu-PSMA-I&T, a different PSMA-targeting ligand, in 11 men with high-risk localized PCa [27]. Patients received two or three doses of Lu-PSMA RLT (7.4 GBq) at 2-week intervals before RP. As in the LuTectomy trial, the treatment was well tolerated with no significant surgical complications. The study reported a median PSA decline of 53.6% and, notably, one patient (9%) achieved a pCR, while 64% had a partial response. Together, these pioneering studies confirm the safety and biological activity of neoadjuvant PSMA RLT but underscore the variability in response, suggesting that achieving high pCR rates will likely require regimen optimization or combination strategies [28].

### 4.3. The Next Frontier in Trial Design: An Overview of Ongoing Studies

The findings from LuTectomy have directly informed the design of the next wave of clinical trials, which seek to build upon its initial observations. These studies, summarized in Table 3, represent the logical evolution of the research agenda in this space.

The NEPI trial (EudraCT 2021-004894-30) is a randomized phase 1/2 study in Germany targeting patients with very high-risk PCa [28]. Responding to the LuTectomy finding that monotherapy is insufficient for pCR, this trial takes the next logical step by evaluating a combination therapy strategy. Patients are randomized to receive 12 weeks of neoadjuvant therapy consisting of ADT plus two cycles of ^177^Lu-PSMA-617, either with or without the addition of ipilimumab, an immune checkpoint inhibitor (ICI) that blocks CTLA-4. The co-primary endpoints are the feasibility of performing RP after this intensive neoadjuvant regimen and, critically, the pCR rate.

## 5. The Immunomodulatory Potential of Neoadjuvant RLT

Beyond its direct cytotoxic effects, RLT has been shown to modulate the tumor microenvironment and elicit systemic immune responses. This “in situ vaccine” effect, whereby targeted tumor cell destruction can transform an immunologically “cold” tumor into a target for the immune system, provides a strong rationale for combining RLT with immunotherapy [62].

### 5.1. Inducing an In Situ Vaccine Effect

Radiation, including that delivered by radionuclides, can induce a form of cell death known as immunogenic cell death (ICD), which is uniquely capable of activating an immune response [63]. Unlike apoptosis, ICD actively signals the immune system about the dying tumor cells. Hallmarks of ICD include the surface expression of damage-associated molecular patterns (DAMPs). Key DAMPs in ICD include the translocation of calreticulin (CRT), which functions as an “eat-me” signal promoting phagocytosis by dendritic cells (DCs) [64]. In addition, dying cells release ATP, which acts as a “find-me” signal to attract DCs to the tumor site [65]. Finally, the release of high-mobility group box 1 (HMGB1) protein promotes DC maturation and the presentation of tumor antigens to T-cells.

This activation of DCs within the tumor can have profound effects on the local microenvironment and drive a systemic anti-tumor T-cell response. Mature DCs present tumor-associated antigens (TAAs) to naïve T-cells in draining lymph nodes, thereby priming tumor-specific cytotoxic CD8^+^ T-cells. These activated T-cells can then traffic back to the primary tumor and to distant metastases to eliminate remaining cancer cells. This concept of the treated tumor serving as the source of antigens for its own vaccination—the in situ vaccine effect—is a central goal of combining radiation with immunotherapy [62].

Figure 2 illustrates the detailed mechanistic cascade initiated by RLT, showing how targeted tumor destruction leads to ICD, antigen presentation by DCs, and the subsequent activation and trafficking of cytotoxic T-cells to trigger a systemic anti-tumor response.

### 5.2. A Hypothesis-Generating Biomarker: The PD-L2 Signature

Identifying which patients are most likely to mount an effective immune response to RLT is a critical step toward personalizing therapy. Recent research has uncovered promising biomarkers rooted in the interferon-gamma (IFN-γ) signaling pathway. For instance, a potentially paradigm-shifting exploratory study analyzed the tumor immune microenvironment of patients with mCRPC undergoing ^177^Lu-PSMA-617 therapy [63,66], correlating baseline tumor gene expression signatures from archival primary tumor tissue with clinical outcomes. The analysis revealed that a higher baseline expression signature for programmed death-ligand 2 (PD-L2) was strongly and significantly associated with a better response to RLT [63]. Patients with a high PD-L2 signature had a much longer median OS compared to those with a low signature (17.2 vs. 5.7 months), with an HR of 0.46. Furthermore, higher PD-L2 expression correlated with a greater PSA decline. Remarkably, the expression of the more widely studied immune checkpoint, PD-L1, showed no significant association with outcomes.

However, this finding must be interpreted with caution. It was a single-center, retrospective, exploratory analysis with a very small sample size for the transcriptomic analysis (*n* = 23) and a long interval between biopsy and RLT. As such, these results are hypothesis-generating and require rigorous validation in larger, prospective cohorts before they can be considered for clinical use.

Despite these limitations, the finding has profound implications. The discovery that the PD-L2 signature may be a dominant predictor of RLT response suggests that radionuclide therapy could induce a distinct immunological phenotype within the tumor microenvironment, one that relies on a different axis of immune regulation than that targeted by most current immunotherapies [67,68,69]. This challenges the default assumption that anti-PD-1/L1 agents are the automatic choice for combination with RLT and provides a strong biological rationale for designing future clinical trials that incorporate agents targeting the PD-L2 pathway or other immune axes, such as the CTLA-4 pathway targeted by ipilimumab in the NEPI trial.

## 6. Optimizing Efficacy: Combination Strategies and Sequencing

The inability of neoadjuvant ^177^Lu-PSMA-617 monotherapy to achieve high pCR rates has prompted the investigation of combination approaches. By integrating RLT with agents that have complementary mechanisms, it may be possible to achieve synergistic anti-tumor effects and overcome resistance [62].

### 6.1. Synergy with Immunotherapy

The combination of RLT with ICIs represents a highly promising investigational strategy. The biological rationale is strong: RLT acts as an “in situ vaccine,” inducing ICD, releasing tumor antigens, and stimulating a T-cell response [62,70]. However, this nascent T-cell response is often rapidly suppressed by immune checkpoint pathways within the tumor microenvironment. ICIs, such as the CTLA-4 inhibitor ipilimumab and PD-1/PD-L1 inhibitors, aim to “release the brakes” on these T-cells, preventing their exhaustion and enabling a more potent and durable systemic anti-cancer response. The NEPI trial, which is evaluating the combination of ^177^Lu-PSMA-617 and ipilimumab, is the first clinical test of this hypothesis in the neoadjuvant PCa setting [28].

### 6.2. Interplay with PARP Inhibitors: Evidence of Cross-Resistance

Poly (ADP-ribose) polymerase (PARP) inhibitors (PARPis) are another class of highly effective agents for a subset of men with mCRPC, specifically those whose tumors harbor mutations in DNA damage repair (DDR) genes such as *BRCA2* [71]. Since both RLT (via radiation) and PARPis (via enzymatic inhibition) ultimately exert their cytotoxic effects by inducing lethal DNA damage, there is a strong biological basis for both potential synergy and cross-resistance [72].

However, the critical question of how to best sequence these two potent therapies remains unresolved. A recent, important retrospective study investigated this question by comparing ^177^Lu-PSMA-617 outcomes in men with DDR-mutated mCRPC who either had previously received a PARPi or were PARPi-naïve [73]. The results were striking. Prior exposure to a PARPi was associated with significantly worse outcomes with subsequent ^177^Lu-PSMA-617. This effect was most pronounced in the subgroup of patients with BRCA2 mutations. In this group, PARPi-naïve patients had a median PSA PFS of 14.0 months, compared to just 2.9 months for those who had been previously treated with a PARPi. The PSA50 response rate was also dramatically lower in the PARPi-exposed group (35% vs. 89%).

This clinical evidence strongly suggests the development of acquired cross-resistance. The underlying biological mechanism is hypothesized to involve potent selective pressure from PARPi therapy, which may enrich the tumor with clones that have overcome the initial DDR defect, for instance, through secondary or reversion mutations in genes like *BRCA2*, or by upregulating alternative DNA repair pathways. These now more robust DNA repair mechanisms would confer resistance not only to PARP inhibition but also to the single- and double-strand DNA breaks induced by ^177^Lu-PSMA-617 radiation. This creates a crucial clinical dilemma: in a patient eligible for both therapies, which should be used first? Using the PARPi first might effectively “burn the bridge” for RLT. This finding has immediate clinical implications for the sequencing of therapies in mCRPC and raises urgent questions for future research, including whether prior RLT induces resistance to subsequent PARPi therapy. Answering this will require prospective, randomized sequencing trials and will be essential for designing future neoadjuvant combination trials, as a significant proportion of men with high-risk localized PCa harbor germline DDR mutations [74]. Furthermore, emerging preclinical data suggest that the concurrent administration of RLT and PARP inhibitors may be synergistic and potentially overcome the acquired resistance observed with sequential therapy, representing a critical direction for future investigation [75].

### 6.3. Modulating the Target: The “PSMA Flare” Phenomenon

A third and highly sophisticated strategy for enhancing RLT efficacy involves manipulating the expression of the target itself. The expression of PSMA on the surface of PCa cells is not static but is dynamically regulated by the androgen receptor (AR) signaling pathway. Paradoxically, inhibiting the AR with ADT or potent AR pathway inhibitors (ARPIs) such as enzalutamide has been shown to temporarily upregulate PSMA gene and protein expression [76,77].

This phenomenon, termed the “PSMA flare,” has been demonstrated in both preclinical xenograft models and clinical imaging studies [77]. In one notable case report, a patient with mCRPC underwent a PSMA PET scan before and four weeks after starting ADT. The post-ADT scan revealed a 7-fold increase in PSMA uptake in known lesions and, remarkably, 13 new metastatic lesions [78]. More recent work showed that a short (9–14 days) course of enzalutamide induced a significant increase (≥20%) in the SUVmax of existing tumor lesions in 56% of cases [76]. This biological insight suggests that PSMA is not a passive biomarker but a therapeutically modifiable target, raising the possibility of using a short course of an ARPI to intentionally induce a PSMA flare immediately prior to PSMA-targeted therapy. This could theoretically increase the absorbed radiation dose delivered to tumors and improve the visualization of low-volume disease on diagnostic scans, potentially making patients with initially low PSMA expression eligible for RLT. Clinical trials are now underway to formally test this hypothesis.

This strategy of combining an ARPI with RLT was prospectively tested in the mCRPC setting in the ENZA-p trial (NCT04419402), a randomized phase 2 trial that compared ^177^Lu-PSMA-617 with or without enzalutamide versus enzalutamide alone [79]. The combination therapy demonstrated a profound improvement in PSA response rate (PSA50 of 84%) and rPFS. Subsequent analysis of secondary endpoints also revealed a significant benefit in OS and quality of life, providing strong clinical evidence that concurrent AR pathway inhibition can potentiate the efficacy of PSMA RLT. These results strongly support the rationale for testing similar combination strategies in the neoadjuvant setting [80].

## 7. Navigating the Path to Clinical Implementation

While the scientific rationale for neoadjuvant RLT is strong and early data are promising, several major hurdles—spanning clinical trial design, biomarker development, and the practical realities of implementing a novel radiopharmaceutical therapy [26]—must be overcome before this approach can be integrated into routine clinical practice.

### 7.1. Validating Endpoints for Accelerated Approval: The Role of MFS

Historically, a major barrier to developing new therapies for localized PCa has been the long timeline required for clinical trials. Because many men with PCa have a long natural history, demonstrating a statistically significant improvement in the gold-standard endpoint of OS can take more than a decade of follow-up [81], which is prohibitive for efficient drug development.

To address this, the Intermediate Clinical Endpoints in Cancer of the Prostate (ICECaP) working group was established to identify and validate a surrogate endpoint for OS [82]. Through a large-scale, patient-level meta-analysis of randomized trials, ICECaP rigorously evaluated MFS—defined as the time from randomization to the first evidence of distant metastasis or death from any cause—as a potential surrogate. The results were convincing. The initial analysis demonstrated a very strong correlation between a treatment’s effect on MFS and its effect on OS, with a trial-level coefficient of determination (R^2^) of 0.92 [83]. A subsequent analysis, ICECaP-2, which included data from more contemporary trials where patients had access to modern systemic therapies for metastatic disease, confirmed that MFS remained a robust surrogate for OS (R^2^ = 0.83) [83].

The validation of MFS as a surrogate for OS is a landmark achievement with profound implications for clinical research in localized PCa. Specifically, it provides regulatory agencies with a statistically sound and clinically meaningful endpoint upon which to base drug approvals, dramatically shortening the required duration of clinical trials. Consequently, MFS is now the accepted primary endpoint for future phase 3 trials of neoadjuvant or adjuvant therapies in high-risk localized PCa and will serve as the basis for the regulatory approval of neoadjuvant RLT.

### 7.2. Identifying MRD by Detecting Circulating Tumor DNA

While MFS provides a validated pathway for drug approval, researchers are actively seeking even earlier biomarkers of treatment efficacy, with one of the most promising being the detection of circulating tumor DNA (ctDNA) to identify MRD [84]. ctDNA consists of small fragments of DNA shed by tumor cells into the bloodstream, carrying the same somatic mutations as the primary tumor. Using highly sensitive next-generation sequencing techniques, researchers can detect these tumor-specific mutations in a patient’s plasma [85].

In numerous other solid tumors, the presence of detectable ctDNA in the blood after curative-intent surgery or therapy is a powerful predictor of subsequent disease relapse, often identifying at-risk patients months or even years before recurrence is visible on conventional imaging [84]. Early feasibility studies in PCa have shown that tumor-informed, personalized ctDNA assays can identify MRD in patients after RP and that the persistence of ctDNA after surgery is associated with a higher risk of subsequent PSA relapse [85]. This approach complements the use of ultrasensitive PSA assays, which can also detect BCR at levels far below traditional cutoffs [86]. The potential application of ctDNA in the neoadjuvant RLT setting is transformative. Clearance of ctDNA from the blood after neoadjuvant RLT and subsequent surgery could serve as a very early and powerful surrogate for treatment success, potentially predicting long-term MFS. Conversely, the persistence of ctDNA could identify patients at the highest risk of relapse, who could then be immediately triaged to receive adjuvant therapy, creating a truly risk-adapted, personalized postoperative management strategy.

The rise of highly sensitive PSMA PET imaging has prompted an alternative treatment paradigm for high-risk disease: preoperative imaging followed by RP and subsequent metastasis-directed therapy (MDT), such as stereotactic body radiotherapy (SBRT), to eradicate any identified oligometastatic lesions [11,87]. While this is a valid strategy for oligometastatic recurrence, its utility in the primary high-risk localized setting must be carefully considered against the neoadjuvant RLT approach.

The “image-and-treat” MDT strategy is limited by its inability to target anything but macroscopic disease visible on a PET scan [87]. Despite its excellent sensitivity compared to conventional imaging, PSMA PET has a detection threshold; it cannot visualize the individual circulating tumor cells or microscopic tumor deposits (<2–3 mm) that define true micrometastatic disease [88]. Therefore, a treatment plan based solely on targeting PET-avid lesions with MDT will invariably fail to address the underlying systemic risk in many high-risk patients [87].

The neoadjuvant RLT paradigm is designed to overcome this very limitation. By systemically delivering a radiopharmaceutical that binds to PSMA-expressing cells, it can treat all disease manifestations simultaneously—both the visible primary tumor and the invisible micrometastases, regardless of their location [89]. It is a proactive, systemic approach aimed at eliminating occult disease before definitive local therapy, thereby addressing the root cause of future metastatic relapse. This positions neoadjuvant RLT as a theoretically more comprehensive strategy for managing the inherently systemic nature of high-risk localized PCa [26].

### 7.3. Key Hurdles in Clinical Adoption: From Efficacy to Global Implementation

While the immediate focus remains on establishing clinical efficacy, a forward-looking perspective requires confronting the complex, long-term hurdles to global implementation. The growing regulatory acceptance of ^177^Lu-PSMA-617 (Pluvicto^®^) for mCRPC provides a foundational framework [90,91,92,93], but its transition to the neoadjuvant setting presents a distinct set of logistical, clinical, and socioeconomic challenges [89].

Successfully scaling neoadjuvant RLT from a late-stage therapy to a first-line option for high-risk localized disease would place unprecedented strain on the healthcare ecosystem. The logistical barriers are formidable, including the need to dramatically expand the global production of radionuclides and radiopharmaceuticals to prevent shortages [94]. Furthermore, a significant infrastructural deficit exists globally; most nuclear medicine departments are not equipped for such a large patient volume, necessitating substantial investment in shielded treatment rooms and specialized equipment. This is compounded by a critical shortage of the highly trained, multidisciplinary personnel—including nuclear medicine physicians, medical physicists, and technologists—required for safe and effective administration [94,95,96,97].

Ethical considerations are paramount, particularly with the development of next-generation α-emitter RLTs. While agents like ^225^Ac offer superior potency, their high-LET radiation causes complex and largely irreparable DNA damage. Clearly, the ethical threshold for accepting the risk of irreversible off-target toxicity to healthy tissues is substantially higher in the curative-intent neoadjuvant setting than in the palliative mCRPC setting [98].

Beyond these practical issues lie profound clinical and ethical challenges, with the foremost among them being the risk of overtreatment and the imperative for stringent patient selection. High-risk localized PCa is a clinically heterogeneous disease; a significant majority of men will not die from their cancer and may be cured with local therapy alone [8,15,99]. Applying a potent systemic therapy like neoadjuvant RLT to all patients in this broad category would inevitably overtreat a substantial number, subjecting them to unnecessary toxicity, financial burden, and potential long-term adverse effects. Therefore, the responsible implementation of this paradigm hinges on the development and validation of robust predictive biomarkers to pinpoint patients with the most aggressive biology who truly require treatment intensification. A multi-faceted approach is essential, integrating: 1) PSMA PET imaging to confirm target expression and stage disease, 2) genomic classifiers (e.g., Decipher) to assess intrinsic tumor aggressiveness, and 3) liquid biopsies like ctDNA to provide direct evidence of micrometastatic disease [100,101,102]. Only through such precise risk stratification can we hope to maximize the benefit for high-risk individuals while minimizing harm to others.

Closely tied to this is the paramount importance of long-term safety and patient-centered, shared decision-making. The ethical threshold for accepting treatment-related toxicity is substantially higher in a curative-intent setting, where long-term quality of life is a primary goal, compared to the palliative mCRPC setting [103]. While early neoadjuvant trials have shown good short-term tolerability [60], the long-term safety profile remains unknown. Data from the pivotal VISION trial in metastatic disease highlight potential risks, including significant hematologic toxicity, persistent xerostomia, and potential nephrotoxicity [25]. The theoretical risk of secondary malignancies, though low with beta-emitters, also cannot be dismissed [89,104]. Consequently, thorough counseling is imperative, ensuring that patients fully understand the balance between the potential for improved oncologic outcomes and the known and unknown long-term risks. Any decision to proceed must be the result of a comprehensive, shared decision-making process. Long-term follow-up from ongoing trials is essential to rigorously quantify these risks and ensure the benefits truly outweigh the potential for lasting harm [26,105].

Finally, the challenges of cost-effectiveness and equitable global access represent a major barrier to widespread adoption. A formal cost-effectiveness analysis of ^177^Lu-PSMA-617 in the mCRPC setting yielded an ICER of $200,708 per QALY gained, a value exceeding common willingness-to-pay thresholds [106]. As this expensive therapy moves into a much larger, earlier-stage population, its high cost threatens to exacerbate healthcare disparities, potentially limiting access to high-income regions and well-resourced centers. Addressing this economic burden will be paramount to ensuring that neoadjuvant RLT, if proven effective, can become an equitably implemented global standard of care rather than a treatment reserved for a select few.

## 8. Conclusion and Future Directions

### 8.1. Summary of the Potential for Neoadjuvant RLT to Reshape the Treatment Paradigm

Neoadjuvant ^177^Lu-PSMA-617 RLT represents a potential paradigm shift in the management of high-risk localized PCa. It challenges the traditional reliance on local therapies for a disease that is frequently systemic from its inception. By delivering a potent, targeted systemic therapy prior to definitive local treatment, this strategy has the multifaceted potential to eradicate occult micrometastases, provide an in vivo readout of treatment sensitivity through pathological response, and prime a systemic anti-tumor immune response. Early data from trials like LuTectomy are encouraging, demonstrating that the approach is safe, feasible, and biologically active. While monotherapy appears insufficient to achieve complete tumor eradication, these initial findings have paved the way for more advanced combination strategies that hold the promise of significantly improving cure rates for men with the most aggressive forms of localized PCa.

### 8.2. Key Unanswered Questions and Roadmap for Future Research

Despite its vast potential, the field of neoadjuvant RLT is still in its infancy, and numerous critical questions remain. Future research must address these uncertainties through a series of meticulously designed prospective trials.

First, the optimal neoadjuvant RLT regimen must be defined. This involves determining the most effective dose, number of cycles, and timing relative to surgery. The LuTectomy trial used a single cycle, the NEPI trial is testing a two-cycle regimen, and future studies may explore further intensification.

Second, the central question of whether a meaningful pCR rate can be achieved must be addressed. This will likely require moving beyond monotherapy. The results of the NEPI trial, combining RLT with immunotherapy, will be a critical first step. Future studies should evaluate other combinations, such as those with PARPis in patients with DDR mutations, or the incorporation of more potent α-emitting radionuclides like ^225^Ac-PSMA.

Third, the optimal sequencing of RLT with other potent systemic agents, particularly PARPis, is essential. The preliminary evidence suggesting cross-resistance underscores the urgent need for randomized trials to determine whether RLT should precede or follow PARPi in eligible patients to maximize cumulative therapeutic benefit.

Fourth, the role of biomarkers in personalizing neoadjuvant RLT needs to be solidified. This includes validating PSMA PET imaging parameters to predict response, confirming the predictive value of immune signatures like PD-L2 to select candidates for immunotherapy combinations, and developing ctDNA-based MRD assays to guide postoperative adjuvant therapy decisions.

Fifth, future trials must address the limitations of the current evidence. The pioneering studies to date, while promising, are characterized by small patient numbers and limited follow-up. Consequently, they have deliberately avoided claims of improving cure rates—an endpoint that itself requires precise definition (e.g., 5- or 10-year MFS). Larger randomized trials with sufficient statistical power and long-term follow-up are imperative to definitively establish a benefit in clinically meaningful outcomes.

Finally, the long-term safety and quality-of-life impacts of neoadjuvant RLT in a curative-intent population must be carefully evaluated. While acute toxicity appears manageable, comprehensive long-term monitoring is crucial to detect any delayed effects, particularly on renal and bone marrow function, especially as more potent α-emitters are introduced.

It is essential to conclude by underscoring the current status of this approach. Despite its strong scientific rationale and encouraging preliminary data, neoadjuvant PSMA RLT is not currently recommended in any clinical guidelines, and it remains an investigational strategy. The conclusions in this review are based on early-phase, hypothesis-generating data, as the field is not yet mature enough to support a systematic review or meta-analysis. Thus, its use should be restricted to the context of well-designed clinical trials approved by institutional and ethical review boards. The path to clinical implementation requires rigorous scientific validation to ensure that this promising new frontier ultimately delivers a true, proven survival benefit to men with high-risk localized PCa—a benefit that has not yet been demonstrated.

## Figures and Tables

**Figure 1 cancers-17-03330-f001:**
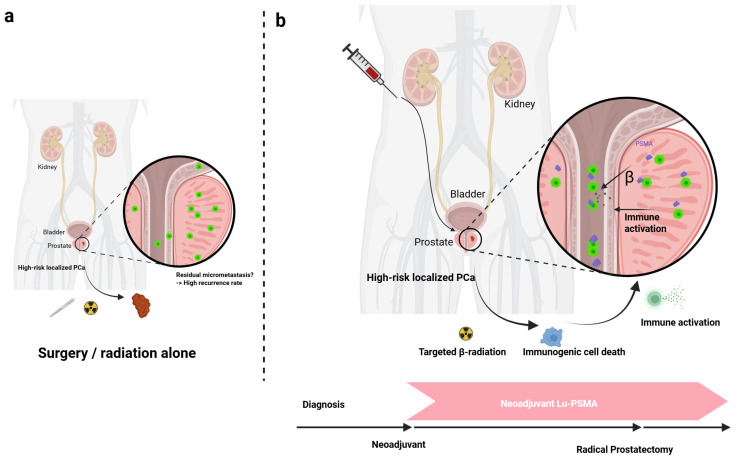
Comparative schematic of treatment strategies for high-risk localized PCa. (**a**) Standard of Care: Definitive local therapy (RP or RT) targets the primary tumor. However, occult micrometastases outside the treatment field are left unaddressed, frequently leading to BCR and distant relapse. (**b**) Neoadjuvant RLT Strategy: Systemically administered ^177^Lu-PSMA-617 acts as a “smart drug,” selectively binding to PSMA-expressing cells. The emitted β-particles deliver targeted radiation to both the primary tumor in the prostate and, crucially, unseen micrometastatic deposits throughout the body. This dual action aims to debulk the primary tumor, treat systemic disease early, and potentially induce a systemic anti-tumor immune response before the planned definitive local therapy. (Figure created using https://BioRender.com/c82u9fm, accessed on 30 July 2025).

**Figure 2 cancers-17-03330-f002:**
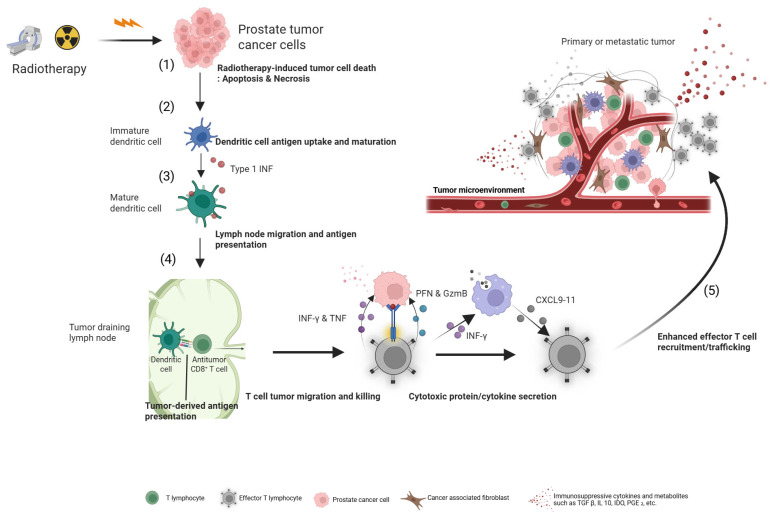
Radio-immunologic modulation of tumor immune microenvironment after ^177^Lu-PSMA-617 RLT. (1) Targeted β-particle radiation from ^177^Lu-PSMA-617 induces ICD in PSMA-expressing tumor cells. (2) Dying cells release DAMPs and TAAs. (3) These signals recruit and activate DCs, which capture the TAAs. (4) The mature DCs migrate to tumor-draining lymph nodes where they present the TAAs to naïve T-cells, priming and activating tumor-specific cytotoxic CD8^+^ T-cells. (5) These activated effector T-cells then traffic back to the tumor and to distant micrometastases, where they recognize and eliminate remaining cancer cells, creating a systemic and durable anti-tumor immune response. (Figure created using https://BioRender.com/c82u9fm, accessed on 30 July 2025).

**Table 1 cancers-17-03330-t001:** Comparison of international guideline definitions for high-risk and very high-risk localized PCa.

Guideline	High-Risk Criteria	Very High-Risk Criteria	Key Features and Differences
NCCN (2025) [7]	One or more of: cT3-cT4 Grade Group 4–5 PSA > 20 ng/mL	Two or more of: cT3-cT4 Grade Group 4–5 PSA > 40 ng/mL	Defines separate “High-Risk” and “Very High-Risk” categories. Uses a higher PSA threshold (40 ng/mL) for the Very High-Risk group.
EAU (2025) [8]	Two or more of: cT3-cT4 Gleason 8–10 (Grade Group 4–5) PSA ≥ 40 ng/mL	Node-positive (N1) disease, OR Meeting the high-risk criteria above	“High-Risk” definition is stricter, requiring two factors and a high PSA. Explicitly includes N1 disease as Very High-Risk.
AUA/ASTRO/SUO (2022) [9]	One or more of: cT3a Grade Group 4–5 PSA > 20 ng/mL	No separate category. Instead, classified as “Locally Advanced Disease”: cT3b-T4 Multiple high-risk features Pelvic-node positivity	Maintains a three-tier system (Low, Intermediate, High). The most advanced cases are termed “Locally Advanced” to emphasize a multimodal treatment approach.

Abbreviations: ASTRO, American Society for Radiation Oncology; AUA, American Urological Association; cT, Clinical T-stage; EAU, European Association of Urology; GG, Grade Group; N1, Node-positive; NCCN, National Comprehensive Cancer Network; PSA, Prostate-Specific Antigen; SUO, Society of Urologic Oncology.

**Table 3 cancers-17-03330-t003:** Summary of key ongoing trials of neoadjuvant PSMA-RLT.

Trial Name (Identifier)	Phase	Patient Population	Intervention(s)	Primary Endpoint(s)	Key Secondary Endpoints
LuTectomy (NCT04430192) [60]	I/II	High-risk localized/locoregional PCa (*n* = 20)	Single cycle of ^177^Lu-PSMA-617 prior to RP	Absorbed radiation dose	Safety, surgical feasibility, PSA response, imaging response, pathological response
NEPI (EudraCT 2021-004894-30) [28]	I/II	Very high-risk localized PCa (*n* = 46)	ADT + 2 cycles of ^177^Lu-PSMA-617 +/- 4 cycles of ipilimumab prior to RP	Feasibility of RP, pCR	Safety, DFS

Abbreviations: ADT, Androgen Deprivation Therapy; DFS, Disease-Free Survival; MFS, Metastasis-Free Survival; OS, Overall Survival; PCa, Prostate Cancer; pCR, pathologic Complete Response; PSA, Prostate-Specific Antigen; PSMA, Prostate-Specific Membrane Antigen; RP, Radical Prostatectomy; SBRT, Stereotactic Body Radiotherapy.

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
