# Peer review of "Neoadjuvant 177Lutetium-PSMA-617 Radioligand Therapy for High-Risk Localized Prostate Cancer: Rationale, Early Clinical Evidence, and Future Directions"

_cancers, 2025, doi:10.3390/cancers17203330_

Round 1

Reviewer 1 Report (Previous Reviewer 3)

Comments and Suggestions for Authors

Thank you for your revision.

Please check redundancy of a description of the following sentences:

“The advent of PSMA-targeted RLT signifies a substantial advancement in the therapeutic management of PCa [30]. This technique hinges on the “theranostic” concept—integrating both therapeutic and diagnostic processes—using one specific molecular target for imaging and treatment, thus allowing for a uniquely customized application of radionuclide therapy. The success of 177Lu-PSMA-617 in the most advanced stages of the disease has provided a robust foundation for its investigation in earlier, curative-intent settings.”

Author Response

Reviewer 2 Report (New Reviewer)

Comments and Suggestions for Authors

The review article titled Neoadjuvant 177Lutetium-PSMA-617 radioligand therapy for high-risk localized prostate cancer: Rationale, early clinical evidence, and future directions provides an overview of neoadjuvant 177Lu-PSMA-617 radioligand therapy for high-risk localized prostate cancer. It brings together available clinical data and explains the biological rationale, therapeutic mechanisms, and practical challenges in an accessible way. The authors present high-risk prostate cancer as an early systemic disease and introduce neoadjuvant radioligand therapy as a potential strategy to address hidden micrometastases before local treatment.

The review is strong in describing international guideline differences, epidemiological trends, and the limitations of current standards of care. It also explains the mechanisms of action, comparing β- and α-emitters, and highlights the theranostic approach. These sections are clear and helpful for both clinical and research audiences.

The discussion of early trials such as LuTectomy and Golan is balanced. The authors emphasize that radioligand therapy appears safe and partly effective, while noting that curative responses have not yet been achieved. They also explore immune effects, biomarker development, and possible drug combinations, drawing from lessons in advanced disease. The manuscript acknowledges logistical, ethical, and economic challenges and suggests pathways for research and equitable access worldwide.

Some limitations are present. Much of the review depends on preliminary data, and there is no systematic quantitative analysis. Long-term safety, cost-effectiveness, and patient selection methods receive limited attention. The reliance on surrogate endpoints such as metastasis-free survival is supported with meta-analyses, but more discussion of overtreatment risk and patient-centered decision-making would be useful. Future directions are outlined well, though the emphasis on drug combinations and sequencing would benefit from comparative trial data and deeper analysis of resistance mechanisms, especially in DNA repair-deficient groups. The authors are recommended to include the recent report on combination therapy of 177Lutetium-PSMA-617 with targeted DNA damage repair inhibitors as discussed in the published abstract https://aacrjournals.org/cancerres/article/85/8_Supplement_1/590/755317

Overall, this review is rigorous and will serve as a foundational resource for neoadjuvant radioligand therapy. It offers valuable insights for both investigators and clinicians but would be strengthened by more detail on the abovementioned points along with methodology, trial limitations, and ethical considerations.

Round 2

Reviewer 2 Report (New Reviewer)

Comments and Suggestions for Authors

The authors have addressed the reviewers’ previous comments with appropriate revisions, reflecting attention to detail and improvement of the manuscript. In its current form, the article meets the standards of quality and scientific rigor required for publication. 

This manuscript is a resubmission of an earlier submission. The following is a list of the peer review reports and author responses from that submission.

Round 1

Reviewer 1 Report

Comments and Suggestions for Authors

Dear authors,

First of all, thank you very much for the opportunity to read and review your extensive work.

I found the review well written and structured, and so easy to read.

I believe, the topic is of greatest interest for the nuclear medicine, potentially leading to a paradigm shift in the prostate cancer therapeutical approach, as correctly stated by the authors.

However, I would suggest to emphasize for the readers, that offering RLT at this stage of the disease, would also mean targeting 2-3 times more patients than what we currently do (currently RLT offered as 2-3 line of treatment in mCRPC). Therefore, I believe they authors should also highlight a bit more extensively the implications of such a shift in the radiopharmaceutical supply, availability of metabolic rooms in nuclear medicine departments, radioprotection measures, and access to this medicine from the global perspective (industrialized countries vs. other regions in the world). In fact, these considerations might be relevant on the long-term for its successful clinical implementation. 

Looking forward to reading the revised version.

Best regards.

Reviewer 2 Report

Comments and Suggestions for Authors

This review article has comprehensively reviewed the importance and future in this field. I think this manuscript can be accepted in the present form.

Reviewer 3 Report

Comments and Suggestions for Authors

This review article focused on standpoint of PSMA RLT nowadays.  Summarizing PSMA RLT trial for mCRPC which have been already reported and discuss about future aspects of using PSMA RLT for neoadjuvant setting of high risk localized prostate cancer.

The review is basically well-written and informative for the readers of the journal.

This reviewer requests some issues for improving the article as the following:

  1. Please summarize PSMA RLT in the global aspect. Which country is accepted for the treatment and which country is now under consideration.
  2. Neoadjuvant ADT before RP for prostate cancer patients did not provide positive results especially in over-all survival. Please discuss about the issue and why neoadjuvant PSMA RLT may bring benefit for prostate cancer patients in comparison to neoadjuvant ADT.
  3. Please discuss neoadjuvant PSMA RLT versus pre-operative PSMA imaging study plus RP plus MDT with or without other systemic therapies.

Reviewer 4 Report

Comments and Suggestions for Authors

The review is extensive and critical. It does not report we have several radioligands in addition to 617, so RLT is more than only 617. A Golan study reported neoadjuvant PSMA I&T. The review does not refer to many RCTs of early phases of mCRPC (Bullseye, SPLASH, ECLIPSE) and especially does not comment of the ENZA-p trial where enzalutamide is combined with PRLT.

The review does not comment on the risk of overtreatment by neoadjuvant therapy.

It is difficult to understand Table 1. Figure 1 and Figure 2. The review should not include PSMA-DC [55].

P16 line 621 Key unanswered questions should add two essential points.  Reported trials had limited number of patients and limited follow up. The reports of the trials wisely abstained from claims of improved cure rates (not defined in the review). At present neoadjuvant therapy with PRLT is not recommended in guidelines and should only be undertaken within the framework of clinical investigations approved by science-ethic committees.

Round 2

Reviewer 1 Report

Comments and Suggestions for Authors

Dear authors,

Thank you for the revised version.

All points were addressed adequately.

No further remarks/requests.

My sincere congratulations.

Best regards.

Reviewer 4 Report

Comments and Suggestions for Authors

Comments to authors
The authors are complimented for rapid response and positive adjustments in revision 1.
Still the review is severely biased. We have two published clinical trials of less then 40 patients with a follow up limited to two years.
The authors argue against neoadjuvant ADT as it does not lead to significant MFS and OS (line 185). At the same time the review states neoadjuvant PRLT is a new frontier (title). But the review does not report MFS or OS benefit for neoadjuvant PRLT (line 720).
The authors report of a series of pioneering trials of neoadjuvant clinical trials of PRLT (line 331) (?).
The Golan trial also included three cycles of PRLT (line 394).
PRLT may make cross firing but that does not reduce heterogeneity of PCa (line 239).
If MFS and OS is the central endpoints the question is not the potential role of PRLT (290) especially as the review argues warmly for combination of PRLT and other drugs.
Please delete “and are vastly different from bets particles (line 307).
The ENZA-p trial also reported OS (line 533)
The review could add that PSA detected with ultrasensitive assays also points to start of BCR before traditional cut off limits (line 581).
The review is concerned of hurdles with global implementation but it is not near foreseen problem (line 607).
“curative intent” is also used for RP and EBRT but leaves a subgroup to progress due to undetected metastases.
The authors may reflect on “localized” is based on conventional imaging where staging with PSMA ET is used increasingly.
I doubt the OS for localized high risk patients is as low as stated. If only 10% die of prostate cancer, 90% will be overtreated with neoadjuvant therapy.
Reference 60 and 61 are identical.
